# Decoupling Primitive with Experts: Dynamic Feature Alignment for Compositional Zero-Shot Learning

**Xiao Zhang**[1,2][*] **Haodong Jing**[1,2][*] **Yongqiang Ma**[1,2][†] **Nanning Zheng**[1,2][†]

[1]National Key Laboratory of Human-Machine Hybrid Augmented Intelligence
[2]Institute of Artificial Intelligence and Robotics, Xi'an Jiaotong University
`{xzhang,jinghd}@stu.xjtu.edu.cn, {musayq,nnzheng}@xjtu.edu.cn`

## Abstract

Compositional Zero-Shot Learning (CZSL) investigates compositional generalization capacity to recognize unknown state-object pairs based on learned primitive concepts. Existing CZSL methods typically derive primitives features through a simple composition-prototype mapping, which is suboptimal for a set of individuals that can be divided into distinct semantic subsets. Moreover, the one-to-all cross-modal primitives matching neglects compositional divergence within identical states or objects, limiting fine-grained image-composition alignment. In this study, we propose **EVA**, a Mixture-of-Experts Framework for Semantic Variant Alignment. Specifically, we introduce **domain-expert adaption**, leveraging multiple experts to achieve token-aware learning and model high-quality primitive representations. To enable accurate compositional generalization, we further present **semantic variant alignment** to select semantically relevant representation for image-primitives matching. Our method significantly outperforms other state-of-the-art CZSL methods on three popular benchmarks in both closed- and open-world settings, demonstrating the efficacy of the proposed insight.

## 1 Introduction

Compositional generalization (Atzmon et al., 2016; Lake et al., 2017) enables artificial intelligence systems to derive new concepts from existing knowledge, thereby accurately interpreting unseen visual instances. Inspired by this capacity for compositional generalization, Compositional Zero-Shot Learning (CZSL) (Misra et al., 2017; Li et al., 2020; Naeem et al., 2021) aims to recognize novel compositions of states and objects by leveraging primitive knowledge acquired from seen compositions. A central challenge of CZSL lies in **primitive polysemy**: the meaning and visual manifestation of the same primitive can differ substantially depending on its compositional context. Addressing this semantic variability is crucial for building robust composition representations.

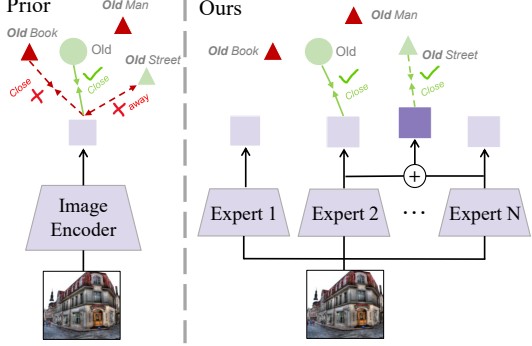

Figure 1: **Left:** Prior primitive alignment neglects compositional divergence within primitive, thus disrupting the topological structure of multimodal space. **Right:** Our method selects the most semantically relevant feature from domain-experts for improved fine-grained cross-modal alignment.

Recent CZSL approaches (Huang et al., 2024; Xu et al., 2024; Lu et al., 2023; Jing et al., 2024) have advanced performance by incorporating composition-level supervision, typically through contrastive objectives that align visual compositions with their textual embeddings. Primitive-level

---

[*]Equal contributions.
[†]Corresponding authors.

alignment, when used, is often introduced as an auxiliary objective to marginally improve performance. Hence, primitive learning has not been sufficiently appreciated and explored; These methods still rely on **single-prototype representations for each primitive**, which are shared across all compositional contexts. This one-to-all constraint disrupts the topological structure of fine-grained primitive-composition relations, leading to semantic entanglement. As a result, the quality of composition embeddings is inherently limited by the underpowered modeling of their primitive constituents. For instance, a single prototype for *young* cannot simultaneously capture the distinct semantics of *young dog* and *young city*, even when composition-level losses are applied. This gap motivates the need for *how to learn primitive features that can dynamically adapt to diverse semantic variants*.

Our work builds on the observation that primitives are not only shared across compositions but are also **semantically heterogeneous**, exhibiting context-dependent variants that cannot be faithfully represented by a single embedding. While prior works (Purushwalkam et al., 2019; Nayak et al., 2022; Lu et al., 2023; Xu et al., 2024) have addressed composition-level alignment, they implicitly assume that primitive embeddings are static and context-invariant, serving as a fixed anchor across all compositions. In Fig. 1, primitive visual features are extracted by a simple expert-module and aligned with single text primitive feature, which fails to extract specific and expressive primitive semantic (*e.g.*, **old** *man* and **old** *street*) and achieve fine-grained cross-modal matching (*i.e.*, different semantic alignment from *old*). Thus, we argue that such simplification restricts the compositional generalization ability of CZSL models, especially in open-world scenarios where primitives frequently combine in unseen ways.

To tackle this issue, we draw inspiration from the Mixture-of-Experts (MoE) paradigm (Zhou et al., 2022; Jacobs et al., 1991; Shazeer et al., 2017; Liu et al., 2024a), which is inherently suited to model the heterogeneous nature of primitive concepts. The core design of MoE—which routes different inputs to specialized experts—aligns perfectly with the fundamental challenge in CZSL: the meaning of a primitive varies dramatically based on its compositional context. Unlike conventional architectures that simply increases model capacity uniformly, MoE enables **dynamic specialization**, allowing the model to capture diverse, context-aware primitive representations, where each expert can specialize in a different semantic facet of a primitive. Importantly, our use of MoE is not merely a transplant of a more powerful architecture; it is a principled, **task-driven design** that leverages expert specialization to explicitly address the semantic variability problem intrinsic to compositional learning. To the best of our knowledge, this work is the first to introduce and motivate the use of MoE for this specific purpose in CZSL.

Based on this insight, we propose **EVA**, a Mixture-of-Experts Framework for Semantic Variant Alignment. **EVA** introduces two effective strategies: *domain-expert adaption* for learning high-quality primitives representations, and *semantic variant alignment* for establishing robust fine-grained image-composition alignment:

1. *Domain-expert adaption* introduces the MoE adapter to process tokens in each layer of the image and text encoders. Through dynamic token allocation, each expert handles semantically similar tokens, facilitating the mastery of **in-domain knowledge** (common knowledge associated with specific domain, *e.g.*, color) and improving performance in prototype semantic modeling. The MoE adapter learns prototypical concepts at token level and integrates composition information through the self-attention layer, ensuring the effective transmission of semantic information. Moreover, considering **knowledge redundancy** in multi-expert collaboration, we designate a shared expert to capture general knowledge, and other activated experts to focus on specialized knowledge. As a result, compared to suffix modules in (Bao et al., 2024; Huang et al., 2024; Xu et al., 2024), **EVA** is an efficient and flexible end-to-end model.

2. *Semantic variant alignment* is the **global-to-local** cross-modal solution from image and text views. In the text domain, since compositions belong to their respective state and object sets, primitive features can be regarded as the centroids of composition features. Thus, without explicitly maintaining a cluster of feature variants, **EVA** captures fine-grained primitive features. In the image domain, `CLS` tokens from various experts are regarded as image feature variants, which are semantic representations of visual content from different perspectives. With the similarity between these variants and their corresponding primitive text features, we select the highest-scoring variants as the primitive visual features. Since explicitly modeling semantic variants of primitives, our method

complements composition-level objectives in prior work and provides a new perspective on how primitives can be structured for robust CZSL.

To evaluate the proposed method, we conduct comparative experiments on three well-known datasets: MIT-States (Isola et al., 2015), UT-Zappos (Yu & Grauman, 2014), and C-GQA (Naeem et al., 2021) in both closed- and open-world settings. Our method significantly outperforms other state-of-the-art CZSL approaches, achieving **+0.7%** and **+2.6%** AUC gains on UT-Zappos and C-GQA in open-world setting. Furthermore, extensive ablation studies robustly demonstrate the effect of the **EVA** components.

## 2 RELATED WORK

**Compositional Zero-shot Learning (CZSL).** CZSL (Misra et al., 2017; Lu et al., 2023; Nayak et al., 2022; Atzmon et al., 2020) learns the entanglement between states and objects from seen compositions in training to recognize unseen compositions during test. Recent methods (Nayak et al., 2022; Xu et al., 2022; Huang et al., 2024; Jing et al., 2024) leverage high-quality features from pre-trained VLM (Radford et al., 2021), achieving impressive zero-shot performance. CSP (Nayak et al., 2022) designs a single learnable prompt for each primitive, without considering the primitive polysemy and the impact of prompt learning on the features is limited. Troika (Huang et al., 2024) proposes a cross-modal traction module to adaptively learn text representations relevant to visual content. However, for various tokens with different semantic contents, single expert module inadequately achieves deep primitives learning. GIPCOL (Xu et al., 2024) introduces a graph-based prompt learning method, but only focuses on the relationships between the primitives at the compositional level. In contrast, we propose *domain-expert adaption* to introduce MoE adapters at each layer of both image and text encoders, enabling deep and efficient intra-primitive feature learning.

**Concept Representation Learning.** Learning transferable and generalizable concept representations is a central challenge in deep learning. In NLP filed, several methods (Vaswani et al., 2017; Devlin et al., 2019; Radford et al., 2019; Brown et al., 2020) attempt to pretrain models on large datasets, learning general and accurate token representations. Recent works (Jiang et al., 2024; Liu et al., 2024a) further utilize the MoE layer to achieve dynamic token routing for in-domain knowledge learning. However, identifying fundamental visual concepts for description remains challenging, visual concept learning typically requires natural language supervision (Li et al., 2021). CLIP (Radford et al., 2021) learns transferable visual representations through contrastive learning on large-scale image-pair datasets. BLIP (Li et al., 2022) is pre-trained with both understanding-based and generation-based tasks to learn generalizable representations. LLaVA (Liu et al., 2024b) introduces visual instruction tuning to align LLM with the multimodal space. These methods focus on learning general visual concept features with the supervision from text, while we propose a multi-expert representation method with the supervision from semantic variants, enhancing primitive features expressiveness for cross-modal alignment.

## 3 METHODOLOGY

### 3.1 PRELIMINARY

The goal of CZSL is to develop a model with compositional generalization capacity, enabling it to accurately recognize unseen compositions based on learned primitive knowledge (*i.e.*, state and object). Given primitive concept set $H$: state label set $\mathcal{S} = \{s_1, s_2, \ldots, s_m\}$ and object label set $\mathcal{O} = \{o_1, o_2, \ldots, o_n\}$, the state-object composition set $\mathcal{C}$ is defined as the Cartesian product of these two primitive concepts sets (*i.e.*, $\mathcal{C} = \mathcal{S} \times \mathcal{O}$). For zero-shot evaluation, we denote seen and unseen composition label sets as $\mathcal{C}^s$ and $\mathcal{C}^u$ respectively, which are disjoint subsets of $C$. The training dataset is defined as $\mathcal{T} = \{(x_i, c_i) | x_i \in \mathcal{X}, c_i \in \mathcal{C}^s\}$, where $\mathcal{X}$ denotes the image space, and only the seen set $\mathcal{C}^s$ is accessible during training. In closed-world setting (Naeem et al., 2021), the test composition set $\mathcal{C}^t$ includes both seen and unseen sets, defined as $\mathcal{C}^t = \mathcal{C}^s \cup \mathcal{C}^u$. In the more challenging open-world setting (Mancini et al., 2021), the state-object composition set $\mathcal{C}$ is utilized as the test composition space.

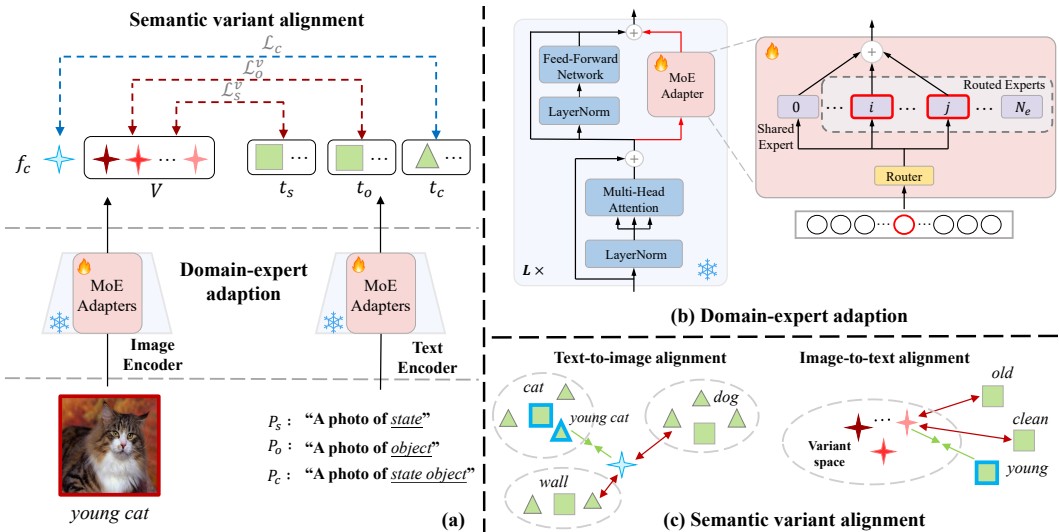

Figure 2: (a). The framework of **EVA** consists of *domain-expert adaption* for token-aware representation learning and *semantic variant alignment* for fine-grained image-primitives matching. (b). *Domain expert adaption* utilizes MoE adapter to dynamically process semantically relevant tokens with in-domain knowledge. (c). *Semantic variant alignment* introduces text-to-image and image-to-text alignment to select most relevant features for cross-modal matching from the text and image views, respectively.

## 3.2 FRAMEWORK OVERVIEW

The human brain consists of different functional areas that work together to complete various tasks. Inspired by distributed functional system, we propose **EVA**, a Mixture-of-**E**xpert Semantic **V**ariant **A**lignment framework for CZSL, which employs several domain-experts for adaptive concept learning and fine-grained semantic alignment, as depicted in Fig. 2 (a). In this study, we adopt the frozen CLIP (Radford et al., 2021) image encoder $E_v$ and text encoder $E_t$ to derive image and text representations, respectively. Specifically, we introduce *domain-expert adaption*, illustrated in Fig. 2 (b), to dynamically process tokens in each layer of encoders. The proposed domain-experts create a sophisticated mapping to deeply learn primitives at the token level. To address existing all-to-one image-primitives issues, we design *semantic variant alignment*, shown in Fig. 2 (c), to select the most semantically relevant feature variants for fine-grained primitive concept learning.

## 3.3 DOMAIN-EXPERT ADAPTION

As depicted in Fig. 2 (b), the MoE adapter, parallel to the Feed-Forward Network (FFN), consists of a router $\mathcal{R}$ for dynamic token allocation and multiple experts $\{\mathcal{E}_i\}_{i=0}^{N_E}$. It is worth noting that we designate a shared expert $\mathcal{E}_0$ to learn common knowledge, while others serve as routed experts $\{\mathcal{E}_i\}_{i=1}^{N_E}$ to focus on domain-specific knowledge. We utilize standard LoRA (Hu et al., 2021) to develop the MoE adapters, where each expert $\mathcal{E}_i$ has two trainable parameters $A \in \mathbb{R}^{r \times d}$, $B \in \mathbb{R}^{d \times r}$ and $r \ll d$. This design makes experts lightweight and prevents overfitting due to the limited size of CZSL datasets. Given the hidden token embedding $h_j \in \mathbb{R}^d$ in layer $j$, the router first computes the token-to-expert affinity $G \in \mathbb{R}^k$ to measure each expert's contribution. Then each token is assigned to shared expert and $K$ routed experts for domain-knowledge learning:

$$G = Softmax(TopK(\mathcal{R}(h_j))), \tag{1}$$

$$\mathcal{E}_i = B_i A_i, \tag{2}$$

$$h_{j+1} = \sum_{i=1}^{N_e} G_i \mathcal{E}_i(h_j) + \mathcal{E}_0(h_j), \tag{3}$$

where $Topk(\cdot)$ function selects the $K$ most relevant experts, while setting the scores of the other experts to $-\infty$. With the implementation of the MoE adapter in both the image and text encoders, our method leverages domain-expert knowledge and inter-expert collaboration to model high-quality

image representation $f_c \in \mathbb{R}^{1 \times d}$ and text representation $t_c \in \mathbb{R}^{1 \times d}$, respectively:

$$f_c = E_v(x), \ t_c = E_t(P_c), \tag{4}$$

where $x \in \mathbb{R}^{h \times w \times 3}$ denotes the input image, and $P_c = [\ \theta_1, ..., \theta_a, \theta_s, \theta_o\ ]$ is the learnable composition prompt, initialized with "a photo of state object". Finally, we compute the composition probability corresponding to the image $x$ as follows:

$$p_c(c|x) = \frac{\exp(f_c \cdot t_c^\top / \tau)}{\sum_{y \in \mathcal{C}^S} \exp(f_c \cdot t_y^\top / \tau)}, \tag{5}$$

where $\tau \in \mathbb{R}$ is the temperature coefficient from pre-trained CLIP. The training objective for composition classification is defined as:

$$\mathcal{L}_c = -\frac{1}{|\mathcal{T}|} \sum_{(x,c) \in \mathcal{T}} \log p_c(c|x). \tag{6}$$

### 3.4 Semantic Variant Alignment

Primitive concept learning is essential for establishing robust and accurate image-composition alignment. Existing methods (Mancini et al., 2021; Lu et al., 2023; Huang et al., 2024) typically align the primitives visual features with state and object text representations, respectively. However, this all-to-one alignment neglects composition divergence within identical primitives, hindering fine-grained composition matching. To address this limitation, as shown in Fig. 2 (c), we propose *semantic variant alignment* which constructs multiple feature variants for adaptive image-primitives relations. This approach performs a global-to-local cross-modal alignment from both image and text perspectives, respectively.

**Text-to-image alignment.** Our variant-based method leverages the local composition distribution to select the most semantically relevant individual, rather than original primitives feature. Specifically, we select the highest matching score among all compositions within state $\hat{s}$ as the image-state matching score $P_s$, formulated as:

$$p_s(\hat{s}|x) = \max_{c_{\hat{s},o}} p_c(c_{\hat{s},o}|x) \cdot \tau_s, \tag{7}$$

where $c_{\hat{s},o} \in C^{target}$ denotes the state $\hat{s}$-relevant composition and $C^{target}$ is composition label space. $\tau_s > 0$ is a trainable coefficient used to adjust the distribution of state probabilities. Similarly, the image-object matching probability $p_o$ can be obtained. To recognize primitive concept $h$, the cross-entropy loss is employed:

$$\mathcal{L}_h = -\frac{1}{|\mathcal{T}|} \sum_{(x,c) \in \mathcal{T}} \log p_h(h|x). \tag{8}$$

where $\mathcal{L}_h$ denotes $\mathcal{L}_s$ and $\mathcal{L}_o$.

**Image-to-text alignment.** The composition-based approach exacerbates the gap between the seen and unseen sets. Therefore, we propose bridging this gap by utilizing constant state and object information to refine the well-structured image representation space during training.

Specifically, given a training image $x$ with state label $s$ and object label $o$, we first model the image feature variants $\{v_i\}_{i=0}^{N_e} \in \mathbb{R}^{N_e \times d}$ from the MoE adapter in the final layer of the image encoder $E_v$. The state feature $t_s \in \mathbb{R}^d$ and object feature $t_o \in \mathbb{R}^d$ are derived from the text encoder $E_t$ based on learnable prompt $P_s$ and $P_o$:

$$t_s = E_t(P_s), \ t_o = E_t(P_o), \tag{9}$$

where $P_s = [\ \theta_1, ..., \theta_a, \theta_s\ ]$ is initialized with "a photo of state", and $P_o = [\ \theta_1, ..., \theta_a, \theta_o\ ]$ is initialized with "a photo of object". Since multiple experts exploit semantic information from various representation subspaces, the output representations denote the feature variants that describe different semantic contents of the input image. Similar to text-to-image alignment, we propose *inter- and intra-model affinity* to select the most semantically relevant variant as the primitive feature.

In terms of *inter-model affinity*, we measure the state affinity score $A_s \in \mathbb{R}^{N_e+1}$ and object affinity score $A_o \in \mathbb{R}^{N_e+1}$ for feature variants $V = \{v_i\}_{i=0}^{N_e}$ and the corresponding text primitive features,

| Method | MIT-States | | | | UT-Zappos | | | | C-GQA | | | |
|---|---|---|---|---|---|---|---|---|---|---|---|---|
| | Unseen ↑ | Seen ↑ | AUC ↑ | HM ↑ | Unseen ↑ | Seen ↑ | AUC ↑ | HM ↑ | Unseen ↑ | Seen ↑ | AUC ↑ | HM ↑ |
| *Closed-World Evaluation* | | | | | | | | | | | | |
| CLIP (Radford et al., 2021) ICML'21 | 46.0 | 30.2 | 11.0 | 26.1 | 49.1 | 15.8 | 5.0 | 15.6 | 25.0 | 7.5 | 1.4 | 8.6 |
| CSP (Nayak et al., 2022) ICLR'23 | 49.9 | 46.6 | 19.4 | 36.3 | 66.2 | 64.2 | 33.0 | 46.6 | 26.8 | 28.8 | 6.2 | 20.5 |
| DFSP (Lu et al., 2023) CVPR'23 | 52.0 | 46.9 | 20.6 | 37.3 | 71.7 | 66.7 | 36.0 | 47.2 | 32.0 | 38.2 | 10.5 | 27.1 |
| GIPCOL (Xu et al., 2024) WACV'24 | 49.6 | 48.5 | 19.9 | 36.6 | 68.5 | 65.0 | 36.2 | 48.8 | 28.4 | 31.9 | 7.1 | 22.5 |
| CDS-CZSL (Li et al., 2024) CVPR'24 | 52.9 | 50.3 | 22.4 | 39.2 | 74.8 | 63.9 | 39.5 | 52.7 | 34.2 | 38.3 | 11.1 | 28.1 |
| Troika (Huang et al., 2024) CVPR'24 | 53.0 | 49.0 | 22.1 | 39.3 | 73.8 | 66.8 | 41.7 | 54.6 | 35.7 | 41.0 | 12.4 | 29.4 |
| PLID (Bao et al., 2024) ECCV'24 | 52.4 | 49.7 | 22.1 | 39.0 | 68.8 | 67.3 | 38.7 | 52.4 | 33.0 | 38.8 | 11.0 | 27.9 |
| RAPR (Jing et al., 2024) AAAI'24 | 53.3 | 50.0 | 22.5 | 39.2 | 72.8 | 69.4 | 44.5 | 56.5 | 36.0 | 45.6 | 14.4 | 32.0 |
| CLUSPRO (Qu et al., 2025) ICLR'25 | 54.0 | **52.1** | 23.8 | 40.7 | 76.0 | 70.7 | 46.6 | 58.5 | 37.8 | 44.3 | 14.9 | 32.8 |
| LOGICZSL (Wu et al., 2025) CVPR'25 | 53.9 | 50.8 | 23.4 | 40.5 | 74.9 | 69.6 | 45.8 | 57.8 | 39.4 | 44.4 | 15.3 | 33.3 |
| **Ours** | **55.0** | 51.2 | **24.0** | 41.0 | **79.6** | 71.2 | **50.2** | 60.2 | **44.6** | **47.1** | **18.8** | **36.9** |
| *Open-World Evaluation* | | | | | | | | | | | | |
| CLIP (Radford et al., 2021) ICML'21 | 14.3 | 30.1 | 3.0 | 12.8 | 20.6 | 15.7 | 2.2 | 11.2 | 4.6 | 7.5 | 0.3 | 4.0 |
| CSP (Nayak et al., 2022) ICLR'23 | 15.7 | 46.3 | 5.7 | 17.4 | 44.1 | 64.1 | 22.7 | 38.9 | 5.2 | 28.7 | 1.2 | 6.9 |
| DFSP (Lu et al., 2023) CVPR'23 | 18.5 | 47.5 | 6.8 | 19.3 | 60.0 | 66.8 | 30.3 | 44.0 | 7.2 | 38.3 | 2.4 | 10.4 |
| GIPCOL (Xu et al., 2024) WACV'24 | 16.0 | 48.5 | 6.3 | 17.9 | 45.0 | 65.0 | 23.5 | 40.1 | 5.5 | 31.6 | 1.3 | 7.3 |
| CDS-CZSL (Li et al., 2024) CVPR'24 | 21.8 | 49.4 | 8.5 | 22.1 | 61.3 | 64.7 | 32.3 | 48.2 | 8.2 | 37.6 | 2.7 | 11.6 |
| Troika (Huang et al., 2024) CVPR'24 | 18.7 | 48.8 | 7.2 | 20.1 | 61.2 | 66.4 | 33.0 | 47.8 | 7.9 | 40.8 | 2.7 | 10.9 |
| PLID (Bao et al., 2024) ECCV'24 | 18.7 | 49.1 | 7.3 | 20.0 | 55.5 | 67.6 | 30.8 | 46.6 | 7.5 | 39.1 | 2.5 | 10.6 |
| RAPR (Jing et al., 2024) AAAI'24 | 20.1 | 49.9 | 8.2 | 21.8 | 59.4 | 69.4 | 33.3 | 47.9 | 11.2 | 45.5 | 4.4 | 14.6 |
| CLUSPRO (Qu et al., 2025) ICLR'25 | 22.1 | **51.2** | 9.3 | **23.0** | 66.2 | 71.0 | 39.5 | 54.1 | 8.3 | 41.6 | 3.0 | 11.6 |
| LOGICZSL (Wu et al., 2025) CVPR'25 | 21.4 | 50.7 | 8.7 | 22.4 | 63.7 | 69.9 | 36.2 | 50.8 | 9.3 | 43.7 | 3.4 | 12.6 |
| **Ours** | **23.2** | 50.8 | **9.4** | 22.8 | **66.5** | 71.6 | **40.2** | **54.2** | **13.3** | **46.9** | **5.6** | **17.9** |

Table 1: **Quantitative results** on MIT-States, UT-Zappos, and C-GQA in Closed- and Open-World setting.

respectively (*i.e.*, $A_s = Vt_s^\top$ and $A_o = Vt_o^\top$). In terms of *intra-model affinity*, we compute the affinity scores $A_v \in \mathbb{R}^{N_e+1}$ for image features and semantic variants, which introduce supervision for the global semantic content (*i.e.*, $A_v = Vf_c^\top$). The affinity score $A_v$ is beneficial for excluding semantic variants that differ significantly from the main semantic content. Furthermore, the overall affinity score $A_S$ and $A_O$ are derived by considering both affinity scores comprehensively:

$$A_S = A_s + \alpha A_v, \ A_O = A_o + \alpha A_v, \tag{10}$$

where $\alpha > 0$ is a balancing coefficient. Finally, we select state image feature $f_s$ from semantic variants $\{v_i\}_{i=0}^{N_e}$ based on the overall affinity scores:

$$f_s = \arg\max_{v_i} a_i^s, \ \{a_i^s \in A_S | 0 <= i <= N_e\}. \tag{11}$$

Similarly, object feature $f_o$ can be obtained. The image-to-text primitives probability is defined as:

$$p_h^v(h|x) = \frac{\exp(f_h \cdot t_h^\top/\tau)}{\sum_{h \in \mathcal{H}} \exp(f_h \cdot t_h^\top/\tau)}, \tag{12}$$

where $p_h^v$ denotes image-to-text state probability $p_s^v$ and object probability $p_o^v$. The training objective of *semantic variant alignment* is defined as:

$$\mathcal{L}_s^v = -\frac{1}{|\mathcal{T}|} \sum_{(x,c) \in \mathcal{T}} \log p_s^v(s|x), \tag{13}$$

$$\mathcal{L}_o^v = -\frac{1}{|\mathcal{T}|} \sum_{(x,c) \in \mathcal{T}} \log p_o^v(o|x). \tag{14}$$

### 3.5 TRAINING AND INFERENCE

**Training objectives.** The final training objective of **EVA** is achieved by optimizing all classification loss functions, formulated as:

$$\mathcal{L} = \mathcal{L}_c + \lambda_1(\mathcal{L}_s + \mathcal{L}_o) + \lambda_2(\mathcal{L}_s^v + \mathcal{L}_o^v), \tag{15}$$

where $\lambda_1$ and $\lambda_2$ are two coefficients. Due to the reliance on label information, *image-to-text alignment* is only applicable during training.

**Inference.** The final composition prediction $\hat{c}_{s,o}$ combines state $p_s$, object $p_o$, and composition scores $p_c$:

$$\hat{c}_{s,o} = \arg\max_{c_{s,o} \in \mathcal{C}^{test}} p_c(c_{s,o}|x) + \beta(p_s(s|x) + p_o(o|x)), \tag{16}$$

where the coefficient $\beta$ is set as 0.5.

| | MIT-States | | | | C-GQA | | | |
|---|---|---|---|---|---|---|---|---|
| | Unseen ↑ | Seen ↑ | AUC ↑ | HM ↑ | Unseen ↑ | Seen ↑ | AUC ↑ | HM ↑ |
| BASELINE | 51.8 | 47.1 | 20.2 | 36.9 | 32.5 | 38.3 | 10.4 | 26.9 |
| Domain-expert Adaption | 54.8 | 50.1 | 23.0 | 40.1 | 42.8 | 45.6 | 17.2 | 35.5 |
| Semantic Variant Alignment | 52.3 | 49.8 | 22.2 | 39.7 | 34.7 | 41.2 | 12.1 | 29.8 |
| Adaption + Alignment | **55.0** | **51.2** | **24.0** | **41.0** | **44.6** | **47.1** | **18.8** | **36.9** |

Table 2: Ablation Studies of core components on MIT-States and C-GQA datasets.

| Expert split | Unseen ↑ | Seen ↑ | AUC ↑ | HM ↑ |
|---|---|---|---|---|
| $0+8$ | 43.5 | 46.5 | 18.0 | 36.2 |
| $1+8$ | **44.6** | **47.1** | **18.8** | **36.9** |
| $2+8$ | 44.2 | 46.8 | 18.4 | 36.2 |
| $4+4$ | 43.9 | 46.2 | 17.8 | 35.9 |

(a) Expert Split

| Variant Alignment | Unseen ↑ | Seen ↑ | AUC ↑ | HM ↑ |
|---|---|---|---|---|
| BASELINE | 42.8 | 45.6 | 17.2 | 35.5 |
| + *t2i alignment* | 43.8 | 46.4 | 18.0 | 36.2 |
| + *inter-model affinity* | 44.2 | 46.8 | 18.5 | 36.5 |
| + *intra-modal affinity* | **44.6** | **47.1** | **18.8** | **36.9** |

(b) Ablation on Semantic Variant Alignment

| Expert Num | Unseen ↑ | Seen ↑ | AUC ↑ | HM ↑ |
|---|---|---|---|---|
| $1+6$ | 44.0 | 46.5 | 18.2 | 36.2 |
| $1+8$ | **44.6** | **47.1** | **18.8** | **36.9** |
| $1+10$ | 44.0 | 46.0 | 18.0 | 36.0 |

(c) Expert Number

| Method | Unseen ↑ | Seen ↑ | AUC ↑ | HM ↑ |
|---|---|---|---|---|
| **EVA** | 44.6 | 47.1 | 18.8 | 36.9 |
| + *semantic isolation* | 44.2 | 46.7 | 18.5 | 36.4 |
| + *load balance* | 43.9 | 46.3 | 18.0 | 36.0 |

(d) Analysis on Domain-expert Adaption

Table 3: Comparison experiments on C-GQA dataset.

# 4 EXPERIMENT

## 4.1 SETTING

**Datasets.** We evaluate the proposed **EVA** on three widely-used CZSL datasets: MIT-States (Isola et al., 2015), UT-Zappos (Yu & Grauman, 2014), and C-GQA (Naeem et al., 2021). MIT-States comprises 53,753 natural images annotated with 115 states, 245 objects, and 1,962 state-object compositions. UT-Zappos contains 50,025 shoes images with 116 fine-grained compositions, including 16 states and 12 objects. C-GQA is the largest CZSL dataset, featuring 39,298 images labeled with 7,767 compositions, encompassing 453 states and 870 objects. In closed-world setting, we follow previous works (Naeem et al., 2021; Lu et al., 2023) to partition datasets into seen and unseen sets.

**Evaluation Metrics.** Following the evaluation protocol established in prior works (Misra et al., 2017; Lu et al., 2023; Huang et al., 2024), we adjust a calibration bias applied to unseen scores from $-\infty$ to $+\infty$, balancing the prediction scores between seen and unseen pairs. We report the Area Under the Curve (**AUC**) and the best Harmonic Mean (**HM**) to quantify the overall performance across seen and unseen compositions. Moreover, we record the best-seen accuracy **Seen** and best-unseen accuracy **Unseen** to assess performance on these two disjoint subsets.

**Implementation Details. EVA** is built upon the pre-trained frozen CLIP (Radford et al., 2021) ViT-L/14 model, which serves as both the image and text encoder. In the *domain-expert adaption* stage, the MoE adapter with a LoRA-based (Hu et al., 2021) intra-layer structure comprises a router $\mathcal{R}$, a shared expert $\mathcal{E}_0$ and routed experts $\{\mathcal{E}\}_{i=1}^{N_e}$. The router $\mathcal{R}$ is a single-layer fully connected network, while the shared and routed experts are two-layer MLPs with identical structures, where the hidden dim $r$ is set to 64. The number of activated experts $K$ is set to 2. Moreover, the coefficient $\alpha$ is set to 0.5 in intra-model affinity. In the final loss function, the weighting parameters $\lambda_1$ and $\lambda_2$ are set to 0.5 and 0.1, respectively.

**Training and Test.** We train **EVA**, implemented in PyTorch (Paszke et al., 2019), using Adam optimizer (Kingma & Ba, 2014) for 20 epochs with a learning rate of $1e-4$. The weight decay is set to $1e-4$, $5e-4$ and $1e-4$ for MIT-States, UT-Zappos, and C-GQA, respectively. Following prior works (Naeem et al., 2021; Lu et al., 2023), we apply post-training calibration to filter out infeasible compositions in the open-world setting during testing.

## 4.2 COMPARISONS WITH SOTAS

We evaluate the quantitative performance of **EVA** in comparison to previous CLIP-based CZSL methods (Radford et al., 2021; Nayak et al., 2022; Lu et al., 2023; Xu et al., 2024; Huang et al., 2024; Li et al., 2024; Bao et al., 2024; Jing et al., 2024; Qu et al., 2025; Wu et al., 2025), in both closed- and open-world settings, as presented in Table 1.

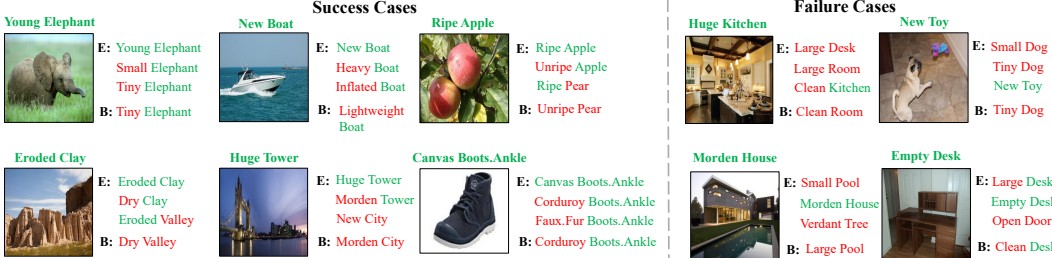

Figure 3: Qualitative Results. We present top-3 predictions from **EVA** (E) and top-1 prediction from BASELINE (B) in terms of success (Left) and failure cases (Right).

**Evaluation in Closed-World Setting. EVA** achieves remarkable improvements over other methods, with AUC gains over LOGICZSL (Wu et al., 2025), improvements of $+0.6\%$, $+4.4\%$, and $+3.5\%$ on MIT-States, UT-Zappos and C-GQA, respectively. Specifically, our **EVA** improves Unseen by at least $+1.0\%$ compared to previous CZSL methods on MIT-States. Notably, our method achieves the highest HM score of $36.9\%$ in C-GQA, demonstrating enhanced compositional generalization compared to other methods (Huang et al., 2024; Li et al., 2024; Jing et al., 2024). These results underscore the advantage of domain-expert adaption in token-aware concept representation learning. In general, our approach **EVA** exhibits robust performance across all datasets and evaluation metrics, consistently outperforming previous state-of-the-art methods, particularly in unseen sets, underscoring its potential for practical applications in closed-world settings.

**Evaluation in Open-World Setting.** Open-world results demonstrate that our method significantly outperforms state-of-the-art approaches, achieving AUC gains over CLUSPRO (Qu et al., 2025), improvements of $+0.1\%$, $+0.7\%$ and $+2.6\%$ on MIT-States, UT-Zappos and C-GQA, respectively. Specifically, our method attains a notable Unseen score of $23.2\%$ on MIT-States dataset, surpassing the previous highest score of $22.1\%$. Additionally, our Seen score of $71.6\%$ exceeds that of several CZSL methods, highlighting a strong balance in handling both seen and unseen compositions. On the most challenging C-GQA dataset, our method surpasses all other CZSL methods across all evaluation metrics, particularly in HM. The improvement of open-world performance demonstrates the effect of semantic variant alignment in establishing accurate image-primitives matching relations. Overall, **EVA** distinguishes itself through its ability to handle unseen data, making it highly effective for open-world tasks where new and unknown compositions are frequently encountered.

## 4.3 ABLATION STUDY

**Effect of Core Components.** We conduct several ablation experiments to access the effect of key components in our **EVA**, as presented in Table 2. The BASELINE employs the frozen CLIP (Radford et al., 2021) encoder without the proposed methods, learning compositional zero-shot capacity with learnable prompts $P_s$, $P_o$ and $P_c$. *Domain-expert adaption* boosts the model's zero-shot performance across all datasets, *e.g.*, improving AUC from $20.2\%$ to $23.0\%$ on MIT-States, indicating that in-domain knowledge learning effectively strengthens primitive semantic modeling. Additionally, we observe that *Semantic variant alignment* leads to a significant improvement, such as an AUC increase from $10.4\%$ to and $12.1\%$ on C-GQA. The integration of both components achieves a new state-of-the-art performance, further demonstrating the effectiveness of our approach.

**Expert Split.** We evaluate the impact of expert split, as depicted in Table 3a. By comparing configurations $0 + 8$ and $1 + 8$, we observe that shared expert enhances compositional recognition performance. However, increasing the number of shared experts leads to a decline in model performance, suggesting that retaining a certain proportion of routed experts is crucial. Furthermore, when the number of shared and routed experts is set to be equal, performance is lower than that of $0 + 8$. This indicates that dynamic routing plays a more significant role than constant collaboration in complex semantic learning.

**Semantic Variant Alignment.** Table 3b provides ablation results for *semantic variant alignment* on C-GQA dataset. Compared to BASELINE without *semantic variant alignment*, text-to-image (t2i) alignment effectively improves compositional zero-shot performance, *e.g.*, $+0.8\%$ improvement in AUC. We observe that progressively incorporating the remaining modules (*i.e.*, inter- and

intra-modal affinity) further increases overall performance. The best results are achieved when all components are utilized, demonstrating the effectiveness of the proposed method.

**Expert Number.** Table 3c demonstrates that the model with 8 experts achieves the highest performance across all metrics, highlighting that a moderate level of expert diversity is most effective for capturing semantic heterogeneity and enabling compositional reasoning. Models with too few experts tend to underfit the complex variations in semantics, failing to disentangle meaningful components, whereas models with an excessive number of experts suffer from routing sparsity, which reduces the effectiveness of each expert and weakens overall specialization. This indicates that balancing the number of experts is crucial for achieving both expressive power and efficient specialization in semantic decomposition.

**Domain-expert Adaption.** To further investigate the impact of semantic isolation on multi-expert learning, we conducted a comparative experiment. Specifically, we employed two identical MoE-based adapters for states and objects, respectively. Each adapter consists of one shared expert and eight routed experts, mirroring the architecture of EVA. We evaluated the models on the most complex dataset, C-GQA, with results summarized in Table 3d.

Introducing semantic isolation leads to a slight performance degradation, suggesting that strictly isolating experts does not improve compositional generalization. This implies that a mixture of attributes and objects may help experts capture semantic relationships, whereas arbitrary divisions offer little benefit. Considering the potential effect of imbalanced expert usage, we further applied a balancing loss to mitigate it. However, even with balanced token loads, performance remains inferior to EVA. This is likely because load balancing enforces uniform expert usage, which contradicts the natural semantic distribution of primitives. Forcing equal routing effectively homogenizes experts, thereby reducing their specialization.

## 4.4 QUALITATIVE ANALYSIS

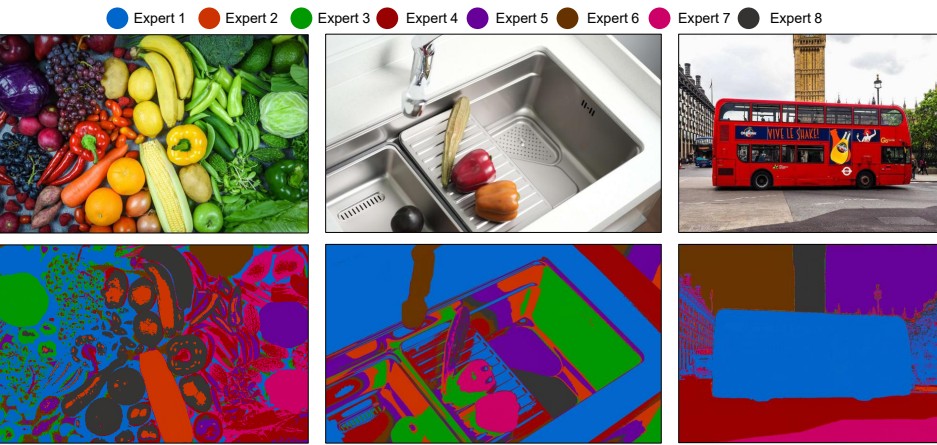

Figure 4: Patch visualization handled by different visual experts.

As illustrated in Fig. 3, we present the success (left) and failure (right) cases of **EVA** and BASELINE, which adopts LoRA-based adapters rather than MoE adapters in both image and text encoders. Compared to BASELINE, **EVA** demonstrates strong compositional recognition capabilities, accurately identifying complex semantic relationships between visual instances, such as *Huge Tower* and *Eroded Clay*. Furthermore, the top-3 predictions are semantically closely related to the Ground Truth, indicating that **EVA** establishes a robust and reliable cross-modal alignment for recognizing compositional relationships between states and objects. In failure cases, while **EVA** and BASELINE both make wrong predictions, **EVA** produces semantically related predictions when interpreting holistic concepts (*Huge Kitchen*) and ambiguous subjects (*Modern House*). This indicates that although **EVA** can handle semantic ambiguity well, its ability to perceive targets in complex visual scenes still requires improvement in future, which may be due to the lack of explicit supervision regarding visual objects.

**Vision Expert Analysis.** In Fig. 4, We observe that different experts attend to distinct semantic regions: some focus on object-level cues (e.g., "bus", "tower"), while others highlight state-related

attributes (e.g., "wet surface," "eroded road"). This specialization validates the intended design of MoE, where experts capture complementary semantic factors. Moreover, experts demonstrate partial redundancy—multiple experts sometimes capture overlapping local regions—which reflects collaborative encoding rather than isolated specialization.

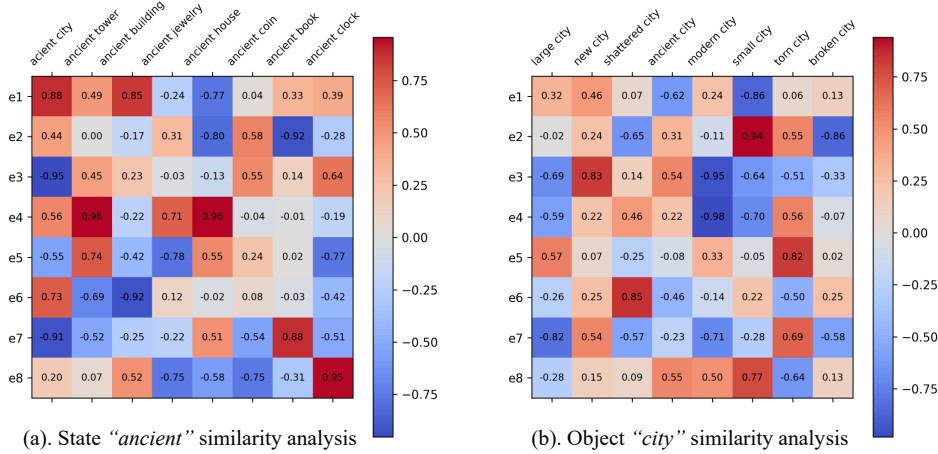

(a). State *"ancient"* similarity analysis    (b). Object *"city"* similarity analysis

Figure 5: Analysis of the similarity between primitive features from different experts and composition features.

**Text Expert Analysis.** Fig. 5 further analyzes the cosine similarity between primitive text features from different experts and composition text features. For example, when evaluating the state *ancient*, certain experts consistently align with *ancient city*, while others align with *ancient library*, showing context-sensitive partitioning of primitive semantics. Similarly, for the object *city*, experts diverge towards *modern city* versus *ancient city*, indicating that experts encode fine-grained semantic variants. This decomposition confirms that **EVA** structurally separates semantic subspaces, mitigating the entanglement present in single-prototype baselines.

Fig. 6 displays the visualization results of attribute and object semantic variants. In terms of state, Semantic variants belonging to different categories are distinguishable. Variant clusters with similar semantic meanings are closer in distance, *e.g.*, large-huge,and the overall distribution of variant clusters corresponds to the correlations of their respective states. The same phenomenon is also observed in the distribution of object variants. Specifically, dog, tiger, and cat all belong to the category of animals, sharing closer proximity to each other, with a dis-

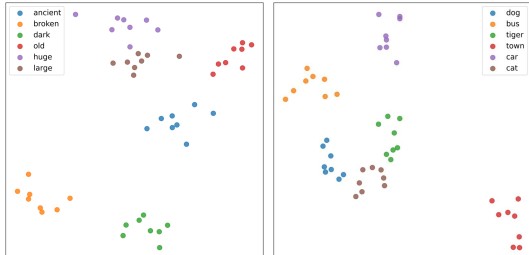

Figure 6: Visualization of semantic variants from text encoder in term of state(***left***) and object(***right***).

tinct boundary separating them from other categories. This indicates that EVA achieves clear semantic separation, with the distribution of semantic variants from different primitives exhibiting semantic plausibility.

## 5 CONCLUSION

In this work, we propose a Mixture-of-Expert Semantic Variant Alignment framework (**EVA**) to address the challenges of concept learning and composition divergence within primitives. Inspired by distributed processing system of the human brain, we leverage MoE adapters to enable an end-to-end model. Through dynamic token allocation, experts specialize as effective in-domain learners, enhancing the modeling of primitive features. Moreover, we introduce *semantic variant alignment* to enable fine-grained and accurate image-primitive mappings. The resulting well-structured primitive representation space facilitates the establishment of discriminative image-composition relations, improving compositional generalization. In future work, we aim to explore strategies to enhance the understanding of abstract concepts and the ability to distinguish object subjects effectively.

ACKNOWLEDGMENTS

This work was supported by the Brain Networks and Brain-Inspired Intelligence Science Breakthrough Pilot Project under Grant No. JYB2025XDXM504 and the Brain Science and Brain-like Intelligence Technology - National Science and Technology Major Project under Grant No. 2022ZD0208800.

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

## APPENDIX

The detailed statistics of the dataset splits are provided in Table 4.

| Dataset | $|S|$ | $|O|$ | $|C|$ | Train $|C^s|$ | Validation $|C^s|$ | $|C^u|$ | Test $|C^s|$ | $|C^u|$ |
|---|---|---|---|---|---|---|---|---|
| MIT-States (Isola et al., 2015) | 115 | 245 | 28175 | 1262 | 300 | 300 | 400 | 400 |
| UT-Zappos (Yu & Grauman, 2014) | 16 | 12 | 192 | 83 | 15 | 15 | 18 | 18 |
| C-GQA (Naeem et al., 2021) | 413 | 674 | 278362 | 5592 | 1252 | 1040 | 888 | 923 |

Table 4: The detailed data split statistics.

**Expert dim $r$.** In Table 5a, we investigate the influence of expert dim $r$ on compositional generalization performance. The optimal dimension is found to be 64, as it achieves the best performance while maintaining training efficiency. When the dimension is gradually reduced below 64, a decline in performance is observed, attributable to information loss caused by dimensional compression. Conversely, increasing the expert dimension to 128 does not yield performance improvements, suggesting that larger dimensions introduce information redundancy.

| Expert dim $r$ | Unseen ↑ | Seen ↑ | AUC ↑ | HM ↑ |
|---|---|---|---|---|
| $r = 8$ | 42.0 | 45.9 | 16.8 | 34.7 |
| $r = 16$ | 42.8 | 46.2 | 17.1 | 35.1 |
| $r = 32$ | 44.2 | 46.8 | 18.2 | 36.5 |
| $r = 64$ | **44.6** | **47.1** | **18.8** | **36.9** |
| $r = 128$ | 43.8 | 46.5 | 18.0 | 36.1 |

(a) Hidden Dim of Expert Adapter

| Number $K$ | Unseen ↑ | Seen ↑ | AUC ↑ | HM ↑ |
|---|---|---|---|---|
| $K = 0$ | 42.4 | 45.3 | 16.7 | 34.7 |
| $K = 1$ | 43.8 | 47.1 | 18.0 | 36.1 |
| $K = 2$ | **44.6** | **47.1** | **18.8** | **36.9** |
| $K = 4$ | 43.2 | 46.9 | 17.8 | 35.9 |
| $K = 8$ | 42.9 | 46.4 | 17.5 | 35.7 |

(b) Activated Expert Number

Table 5: Ablation experiments on C-GQA dataset.

**Activated Expert Number.** Table 5b reports the impact of activated expert number $K$. The optimal $K$ is 2, which achieves a win-win situation in terms of performance and computational cost. When we reduce $K$ to 0, *i.e.*, only utilizing the sharing expert, the MoE adapter becomes a MLP-based adapter, resulting in a performance decline. We next set one activated expert and observe the improvement of CZSL performance, which suggests that the dynamic token routing is beneficial to primitives representation modeling. Larger number $K$ causes the lower performance, indicating the low efficiency in collaboration among multiple experts.

**Analysis of hyperparameter $\alpha$.** The coefficient $\alpha$ is introduced to balance inter- and intra-model affinity. Analogous to weighting terms in loss functions, it enables a comprehensive consideration of both types of affinity when selecting appropriate feature variants. The sensitivity analysis results are presented in Table 6. Across the range from $\alpha = 0$ to $\alpha = 1$, all metrics vary only slightly,

| $\alpha$ | Unseen ↑ | Seen ↑ | AUC ↑ | HM ↑ |
|---|---|---|---|---|
| 0 | 43.8 | 46.2 | 18.0 | 35.8 |
| 0.3 | 44.2 | 46.9 | 18.6 | 36.6 |
| 0.5 | 44.6 | 47.1 | 18.8 | 36.9 |
| 0.7 | 44.0 | 46.8 | 18.5 | 36.3 |
| 1.0 | 43.8 | 46.8 | 18.2 | 36.0 |

Table 6: The balancing coefficient $\alpha$

indicating that EVA is largely insensitive to the exact choice of $\alpha$. Performance remains stable over a broad interval, suggesting that the model is robust to this hyperparameter.

**Effect of Semantic Variant Alignment.** SVA is proposed to address semantic divergence and achieve fine-grained image-primitive alignment. Table 7 shows that Semantic Variants Alignment (SVA) achieves higher primitive prediction accuracy and AUC than BASELINE. It demonstrates that our variants-based primitive alignment is more effective than previous single-prototype method.

| Method | MIT-States | | | C-GQA | | |
|---|---|---|---|---|---|---|
| | att | obj | AUC | att | obj | AUC |
| BASELINE | 41.0 | 50.0 | 23.0 | 48.5 | 60.0 | 17.2 |
| +SVA | **42.7** | **52.1** | **24.0** | **50.0** | **63.1** | **18.8** |

Table 7: Ablation on Semantic Variants Alignment

**Efficiency Comparison.** Additionally, Table 8 shows the efficiency comparison on UT-Zappos dataset. Since we utilize an efficient strategy to design MoE adapters and primitives alignment method, our model achieves superior performance and inference speed without massive trainable parameters.

| Method | #Params(M) ↓ | Training Time ↓ | Inference Time ↓ | AUC ↑ |
|---|---|---|---|---|
| RAPR (Jing et al., 2024) | 37.9M | 3min32s | 28.4ms | 44.5 |
| Ours | **36.7M** | **3min30s** | **12.0ms** | **50.2** |

Table 8: Efficiency comparison with SoTA method.

**Design Comparison.** Table 9 further demonstrates the effect of our design, where suffix module has same structure with MoE adapter. Compared to independent suffix modules, the LoRA-based intra-layer adapter enables a flexible end-to-end model with higher AUC and HM.

| Method | UT-Zappos | | C-GQA | |
|---|---|---|---|---|
| | AUC | HM | AUC | HM |
| Suffix Module | 45.2 | 52.7 | 15.8 | 32.8 |
| LoRA (Ours) | **50.2** | **60.2** | **18.8** | **36.9** |

Table 9: Design comparison.

**Analysis of hyperparameter sensitivity.** We further study the sensitivity of hyperparameter $\lambda_1$ and $\lambda_2$ for loss functions, and $\beta$ for composition inference. In Fig. 7, as the parameters values change, the fluctuation range of the AUC remains within $1\%$, which validates the robustness of our method.

**Token load.** To evaluate the impact of MoE adapter on in-domain knowledge learning, we analyze the computational load of each expert in learning state (Fig. 8 (a)) and object (Fig. 8 (b)). We observe an imbalanced distribution of token processing across experts in both state and object domains. In the state domain, experts $\mathcal{E}_4$ and $\mathcal{E}_6$ process the majority of tokens, while the remaining experts handle a similar token load. In the object domain, experts $\mathcal{E}_1$ and $\mathcal{E}_7$ exhibit the highest computational

load. Since a single text encoder is utilized, this imbalance suggests knowledge separation, with certain experts specializing in specific domains, $i.e.$, expert $\mathcal{E}_4$ excels in state-related tasks, while expert $\mathcal{E}_1$ performs well in object-related tasks.

**Visualization of variant alignment.** Fig. 9 demonstrates that primitive visual variants extracted from different expert models encode diverse semantic information. For instance, V1 and V3 (local) highly relevant to primitive-level semantics and therefore are more suitable for primitive alignment, while image-level features (global) aggregate high-level contextual information across the entire scene, resulting in a loss of fine spatial details that are critical for distinguishing primitive object or object-specific attributes.

**Feature Analysis.** Fig. 10 visualizes the image features learned by BASELINE and **EVA**. Leveraging *domain-expert adaption* and *semantic variant alignment*, **EVA** constructs a well-structured representation space, where features corresponding to identical states or objects are more tightly clustered, and class boundaries are more distinct. This structured representation enhances compositional generalization to unseen instances.

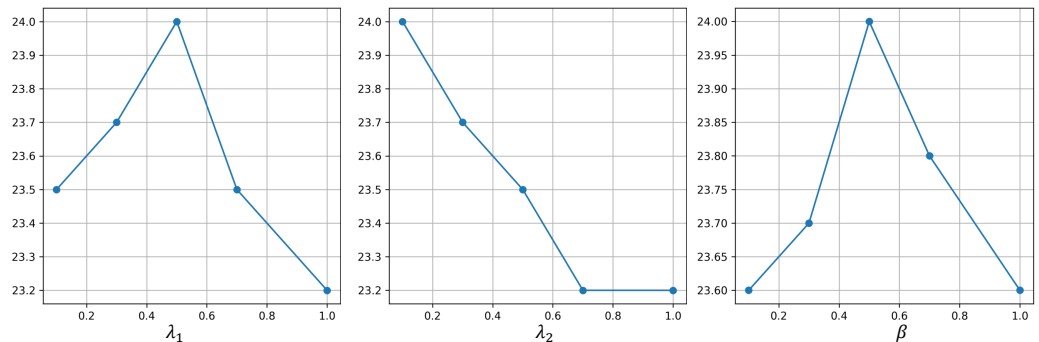

Figure 7: Hyperparameter sensitivity analysis on MIT-States dataset in terms of AUC.

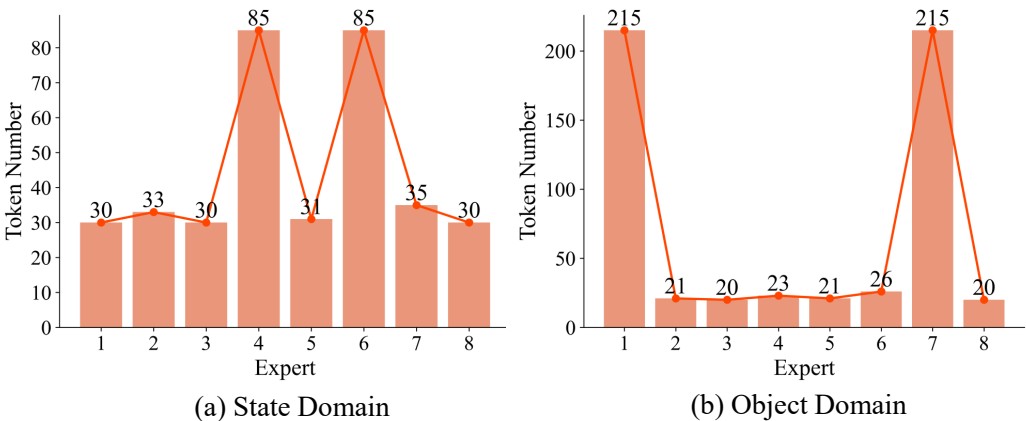

Figure 8: The token load of various experts in state and object domains.

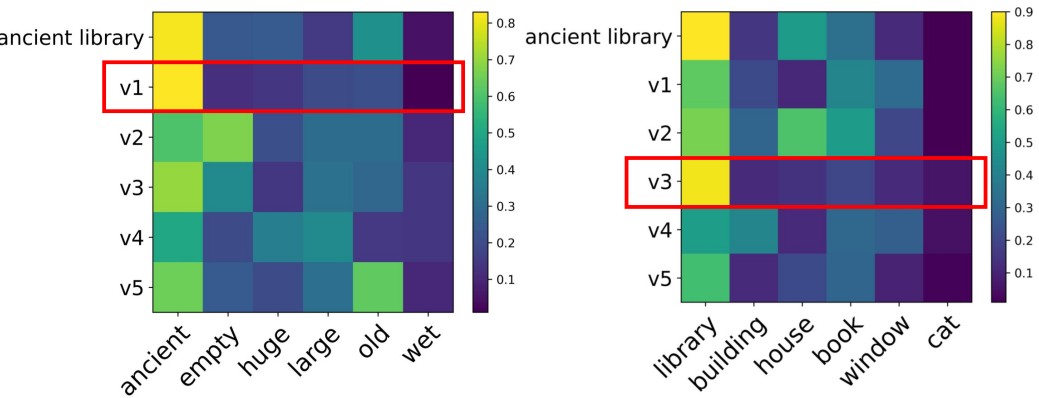

Figure 9: Visualization of the affinity between images feature "ancient library" and variations (v1,..., v5) with states and objects.

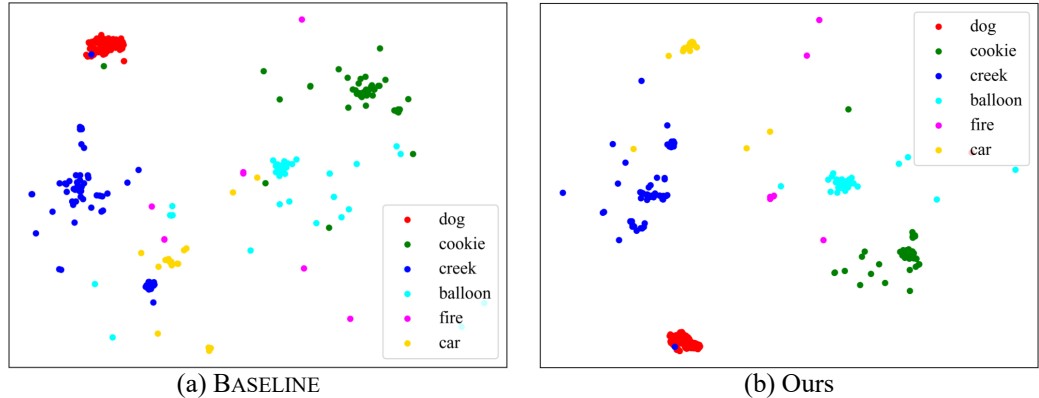

(a) BASELINE          (b) Ours

Figure 10: Visualization of image features learned by BASELINE and our method.

