# OpenReview forum: "Decoupling Primitive with Experts: Dynamic Feature Alignment for Compositional Zero-Shot Learning"
_ICLR.cc/2026/Conference — ICLR 2026 Poster_

### Official Review · Reviewer_nDeF · 2025-10-27

**Soundness:** 2
**Presentation:** 3
**Contribution:** 2
**Rating:** 4
**Confidence:** 3

**Summary:**

This paper studies the Compositional Zero-Shot Learning (CZSL) problem, which aims to recognize unseen combinations of state and object primitives.
Existing methods usually assume a single static prototype for each primitive (e.g., old, young), ignoring the semantic polysemy that arises across different contexts (e.g., old man vs. old book). This leads to semantic entanglement and limited generalization.
To address this, the authors propose EVA (Expert-based Variant Alignment), a Mixture-of-Experts framework that explicitly models semantic variability at the primitive level.
It consists of two main components:
(1) Domain-Expert Adaption, which inserts lightweight LoRA-based MoE adapters into a frozen CLIP backbone to dynamically route tokens to specialized experts, enabling context-aware primitive representations; and
(2) Semantic Variant Alignment, which selects semantically relevant feature variants from both text and image perspectives to achieve fine-grained cross-modal matching.
Experiments on MIT-States, UT-Zappos, and C-GQA datasets demonstrate that EVA consistently outperforms state-of-the-art methods in both closed- and open-world settings, achieving up to +3.5% (closed-world) and +2.2% (open-world) AUC improvements while remaining efficient and interpretable.

**Strengths:**

1. Clear motivation addressing primitive polysemy in CZSL.
2. Strong empirical results across three benchmarks.
3. Lightweight and interpretable design built on frozen CLIP.

**Weaknesses:**

1. The paper lacks a deeper analysis of why MoE improves compositional classification — explanations remain intuitive without quantitative evidence of expert complementarity or routing diversity.
2. The observed gain may stem from implicit feature reparameterization rather than genuine semantic disentanglement.
3. No study of routing stability or expert utilization balance; potential risk of expert collapse.

**Questions:**

1.	Are the experts activated evenly during training, or does routing collapse occur?
2.	Can the authors provide quantitative or visual evidence showing that different experts learn complementary semantics?
3.	Would EVA still work without explicit state/object labels in open-vocabulary or unsupervised settings?
4.	How consistent are the improvements across different vision–language backbones (e.g., BLIP, SigLIP)?
5.	Does the MoE module truly enhance semantic reasoning, or mainly increase representational capacity?

---

> ### Author Response · Authors · 2025-11-24
>
> We thank the reviewer for the constructive feedback and for acknowledging the motivation, empirical performance, and lightweight design of EVA. Below we address the concerns regarding expert behavior, semantic disentanglement, and generalization.
>
> **Q1. Expert learns complementary semantics.**
>
> **A1.** Thanks for your comment. We provide the Fig 4 and 5 to demonstrate visual and text experts indeed learn complementary semantics, respectively. In appendix Fig 10, we additionally visualize the semantic variants across experts. This shows that these variations are not simple random perturbations, but rather capture the fine-grained semantic differences of primitives in different combinatorial contexts.
>
> Specifically, Figure 4 shows that visual experts focus on complementary local regions in an image (e.g., object structure vs. state texture), while Figure 5 confirms that textual experts can dynamically route ambiguous primitives (e.g., 'City') to different feature subspaces based on combinatorial context (e.g., 'Ancient City' vs. 'Modern City'). This complementary expert division of labor and dynamic routing mechanism effectively decouples complex semantic entanglement, thereby significantly improving the performance of component classification.
>
> &nbsp;
>
> **Q2.  Reparameterization rather than semantic disentanglement.**
>
> **A2.**. In the Main text Fig 4, different regions of the image are processed by different experts, exhibiting clear semantic boundaries. This indicates that visual experts have learned semantic differences. In the Main text Fig 5, variants from text experts show different affinities with compositions in terms of state and object. This phenomenon indicates that semantic variants indeed capture the different semantic contents of primitives. Therefore, the proposed EVA goes beyond simple reparameterization.
>
> &nbsp;
>
> **Q3. Open-vocabulary or label-free setting.**
>
> **A3.** In this work, we introduce the domain-expert adaption and semantic variant alignment to learn the fine-grained cross-modal matching for CZSL.
>
> Specifically, 1. Domain-expert adaption is a structural dynamic separation used to learn the diverse semantic within primitives. This can be combined with open-vocabulary or label-free methods. 2. Semantic variant alignment aims to select the most semantically relevant variant for image-primitive alignment, which relies on specific state and object information. Therefore, it cannot work with explicit state/object labels in the label-free setting.
>
> &nbsp;
>
> ​**Q4. Is routing stable? Does expert collapse occur?**
>
> **A4.** Yes, from the perspective of token load, the experts are unbalanced and we didn’t utilize the load balancing loss. There are primarily two reasons for this:
>
> (1). The purpose of introducing dynamic expert learning is to achieve semantic differentiation, and artificially balancing the load would contradict this intention. This reflects **natural specialization** rather than problematic imbalance, as shown in Fig. 4 and 5.
>
> (2). Load balancing encourages experts to be used uniformly, which contradicts the natural semantic distribution of primitives: Primitive semantics are not uniformly distributed across contexts (“old” appears more often with people than fruits), and forcing equal expert routing pushes EVA toward homogenized experts, reducing specialization.
>
> To validate it, we designed a controlled experiment, and the results are as follows:
>
> | Method | U | S | AUC | HM |
> | :--- | :--- | :--- | :--- | :--- |
> | **Eva** | 44.6 | 47.1 | 18.8 | 36.9 |
> | **+Load Balance** | 44.2 | 46.8 | 18.5 | 36.4 |
>
> We observe a consistent but small performance drop when applying a load-balancing (LB) loss.
>
> &nbsp;
>
> **Q5. Improvements across different vision–language backbones.**
>
> **A5.** EVA benefits from CLIP’s cross-modal alignment but is not structurally tied to it. MoE routing and SVA variant selection only require: two Transformer encoders and token-level hidden states and text–image embeddings. To verify the effect of our method, we replace a pretrained **SigLIP-B-16-224** as the backbone while keeping the other settings unchanged.
>
> In the closed-world setting, the results on MIT-States dataset are as follows:
>
> | Method | U | S | AUC | HM |
> | :--- | :--- | :--- | :--- | :--- |
> | **BASELINE** | 53.5 | 49.8 | 23.0 | 40.5 |
> | **+EVA** | 57.9 | 52.1 | 26.4 | 43.6 |
>
> The improvements over the BASELINE—especially the +4.4 unseen accuracy and +3.4 AUC. It demonstrates that EVA provides a consistent improvement.
>
> &nbsp;
>
> **Q6. Does MoE truly enhance semantic reasoning?**
>
> **A6.** By answering the above questions, it shows that our approach is not merely simple feature learning, but rather facilitates expert learning of primitive differences through semantic variant alignment to achieve fine-grained compositional semantic reasoning.

---

> ### Author Response · Authors · 2025-11-27
>
> Dear Reviewer nDeF,
>
> We greatly value your feedback and suggestions, particularly regarding the in-depth analysis of MOE and the further discussion on the generalization of our methods. Your recommendations have enabled us to significantly enhance this work, resulting in improved presentation and stronger persuasiveness.
>
> We are confident that your exceptional expertise and dedication will further validate this work. We kindly request your time to review our rebuttal, which we believe addresses all your concerns. We look forward to further discussion. Wishing you all the best!

---

### Official Review · Reviewer_Fu3k · 2025-10-30

**Soundness:** 2
**Presentation:** 2
**Contribution:** 2
**Rating:** 6
**Confidence:** 5

**Summary:**

This paper addresses the problem of primitive polysemy in Compositional Zero-Shot Learning (CZSL) and proposes an expert-based dynamic feature decoupling framework called EVA (Expert-based Variant Alignment). The authors point out that primitives can have different meanings under different semantic contexts (e.g., “old” in “old man” vs. “old building”), which are difficult to model using a single embedding. As a result, traditional attribute–object composition approaches cannot fully capture semantic variations.

EVA mainly consists of two core modules:

1.	Domain Expert Adaptation: Inserts a Mixture-of-Experts (MoE) module into the CLIP encoder. Through a token-level routing mechanism, it selects the appropriate expert to model semantic variations across contexts.

2.	Semantic Variant Alignment: Designs a cross-modal dynamic alignment strategy that combines global and local feature matching to achieve fine-grained consistency between visual and textual semantic spaces.

Experiments on standard CZSL datasets — MIT-States, UT-Zappos, and C-GQA — show that EVA outperforms several recent methods (e.g., GIPCOL, Troika, IVLP) under both closed-set and open-set settings. The authors claim EVA achieves more stable semantic generalization and better interpretability.

**Strengths:**

1.	The paper proposes a logically coherent and complete MoE + alignment framework, effectively integrating expert modeling with semantic disentanglement.

2.	Experiments cover multiple mainstream CZSL datasets, and the reported performance exceeds several existing methods on certain metrics, showing credible results.

**Weaknesses:**

1. The work is essentially an architectural extension of existing CLIP-based compositional alignment methods.
2. The core mechanisms (expert routing and alignment strategy) lack quantitative validation.
3. The paper does not analyze computational cost, parameter scale, or inference latency, making it difficult to assess practicality.
4. Some figures and text repeat information, reducing clarity.
5. Differences between variants are small, making it unclear how much each module contributes.

**Questions:**

1. Does the expert module experience imbalance during training? Have you introduced a load balancing loss?

2. What is the performance drop if the alignment module is removed or only the MoE structure is retained?

3. Does the EVA framework rely on CLIP’s pretrained semantic structure? Would it still work with non-CLIP backbones?

4. Can you provide more interpretability results (e.g., clustering distributions of experts across semantic attributes) to demonstrate the effectiveness of semantic decoupling?

---

> ### Author Response · Authors · 2025-11-24
>
> We thank the reviewer for the thoughtful assessment and detailed comments.
>
> **Q1. Novelty.**
>
> **A1.** Sorry for this confusion. Please allow us to clarify that EVA is not a simple architectural extension of CLIP but introduces a new modeling principle for CZSL.
>
> Existing CZSL methods operate within a ​**single latent space**​, adjusting a single primitive embedding shared across all compositions. In this paper, we propose a novel insight to reformulate primitive polysemy as semantic heterogeneity and introduces an **architectural decomposition** where multiple experts encode distinct semantic subspaces of the same primitive, enabling context-aware semantic variant modeling. Furthermore, our **semantic variant alignment** is a new cross-modal mechanism: it considers semantic polysemy and performs bidirectional global-to-local variant selection on both text and image sides, rather than the one-to-all primitive alignment used in prior CZSL works. This introduces a new alignment paradigm and yields substantial gains, far exceeding typical incremental improvements in CZSL.
>
> &nbsp;
>
> **Q2. The core mechanisms (expert routing and alignment strategy) lack quantitative validation & performance drop?**
>
> **A2.** Thanks for your review. In the main text table 2, we report the ablation results on expert routing and alignment strategy. These results demonstrate the effectiveness of the proposed method.
>
> &nbsp;
>
> **Q3. Computational cost, parameter scale, or inference latency.**
>
> **A3.** Thanks for your review. In the Appendix, Table 8 reports the efficiency comparison on UT-Zappos dataset. To further evaluate our method on the large dataset, we report the efficiency comparison on MIT-States dataset in the closed-world setting:
>
> | Model | #Trainable | Training Time (per epoch) | Inference Time | AUC |
> | :--- | :--- | :--- | :--- | :--- |
> | **DFSP** | 43.3M | 3min45s | 17.2ms | 20.6 |
> | **Ours** | 41.7M | 3min20s | 17ms | 24.0 |
>
> These two tables shows our methods achieve superior balance between performance and computational cost.
>
> &nbsp;
>
> **Q4. Some figures and text repeat information, reducing clarity.**
>
> **A4.** We appreciate this feedback. We have reorganized the figures and revised overlapping descriptions in Sections 3.1–3.5 to improve clarity.
>
> &nbsp;
>
> **Q5. Expert imbalance & load balancing.**
>
> **A5.** Yes, from the perspective of token load, the experts are unbalanced, and we didn’t utilize the load balancing loss. We think that MoE routing imbalance in EVA is not a defect but a reflection of the intrinsic heterogeneity of primitive semantics. There are primarily two reasons for this:
>
> (1). The purpose of introducing expert learning is to achieve semantic differentiation, and artificially balancing the load would contradict this intention. This reflects **natural specialization** rather than problematic imbalance, as shown in Fig. 4 and 5.
>
> (2). Load balancing encourages experts to be used uniformly, which contradicts the natural semantic distribution of primitives: Primitive semantics are not uniformly distributed across contexts (“old” appears more often with people than fruits), and forcing equal expert routing pushes EVA toward homogenized experts, reducing specialization.
>
> To validate it, we designed a controlled experiment, and the results are as follows:
>
> | Method | U | S | AUC | HM |
> | :--- | :--- | :--- | :--- | :--- |
> | **Eva** | 44.6 | 47.1 | 18.8 | 36.9 |
> | **+Load Balance** | 44.2 | 46.8 | 18.5 | 36.4 |
>
> We observe a consistent but small performance drop when applying a load-balancing (LB) loss.
>
> &nbsp;
>
> **Q6. Dependence on CLIP backbone.**
>
> **A6.** EVA benefits from CLIP’s cross-modal alignment but is not structurally tied to it. MoE routing and SVA variant selection only require: two Transformer encoders and token-level hidden states and text–image embeddings. To verify the effect of our method, we replace a pretrained **SigLIP-B-16-224** as the backbone while keeping the other settings unchanged.
>
> In the closed-world setting, the results on MIT-States dataset are as follows:
>
> | Method | U | S | AUC | HM |
> | :--- | :--- | :--- | :--- | :--- |
> | **BASELINE** | 53.5 | 49.8 | 23.0 | 40.5 |
> | **+EVA** | 57.9 | 52.1 | 26.4 | 43.6 |
>
> The improvements over the BASELINE—especially the +4.4 unseen accuracy and +3.4 AUC. It demonstrates that EVA provides a consistent improvement.
>
> &nbsp;
>
> **Q7. Request for more interpretability.**
>
> **A7.** Thanks for your suggestion. In the revision version, we have provided more results and analysis for interpretability.

---

> ### Author Response · Authors · 2025-11-27
>
> Dear Reviewer Fu3k,
>
> We are deeply grateful for your recognition of our work. In our rebuttal, we have focused on discussing the innovation of our methodology and its robust performance. We believe you are an outstanding expert in this field with profound knowledge, and your comments on this paper are well-considered. The questions you raised are highly beneficial for the long-term development of this paper, which is precisely why we have worked so diligently.
>
> We hope you will take the time to review our rebuttal, which we believe addresses all your concerns. We look forward to further discussion. Wishing you all the best!

---

### Official Review · Reviewer_BePU · 2025-10-31

**Soundness:** 4
**Presentation:** 3
**Contribution:** 3
**Rating:** 6
**Confidence:** 5

**Summary:**

This paper addresses Compositional Zero-Shot Learning (CZSL) focusing on the challenge that primitive features (states, objects) vary semantically across contexts. The authors propose EVA (Expert-based Variant Alignment), a MOE framework that learns context-aware primitive representations and performs fine-grained cross-modal alignment with carefully designed traning objective. EVA introduces two key components:(1) Domain-Expert Adaption, which employs MoE adapters in image and text encoders to capture diverse semantic facets of primitives through dynamic token routing; and(2) Semantic Variant Alignment (SVA), which aligns fine-grained visual and textual variants via both text-to-image and image-to-text matching. Comprehensive quantitative and qualitative experiments on the three most commonly used CZSL datasets  demonstrate that EVA achieves superior performance  in both closed- and open-world settings. The analyses further show that different experts specialize in distinct semantic aspects, validating the effectiveness of the proposed design.

**Strengths:**

1. The paper introduces an MoE-based framework to address the semantically heterogeneous of primitives in CZSL. By dynamically routing tokens to domain-specific experts, the proposed approach effectively captures context-dependent primitive semantics.The proposed model achieves SOTA performance in both closed-world and open-world settings without any suffix modules.


2. The proposed SVA module overcomes the limitation of previous all-to-one alignment  by introducing a global-to-local cross-modal alignment from both image and text perspectives. This design enables more accurate and context-sensitive alignment of visual and textual primitives, leading to better compositional discrimination.

3. The paper is well-structured and easy to follow, with thorough quantitative and qualitative evaluations.

**Weaknesses:**

1.  Figure 2(c) does not clearly illustrate how the SVA module works, and Figure 2(a) lacks explicit description for the text-to-image alignment pathway $\mathcal{L}_s^v ,\mathcal{L}_o^v$.
2. The paper does not include an ablation study on the number of experts.
3. The paper lacks deeper analysis of expert specialization, such as overall statistical distributions or further exploration of text experts.

**Questions:**

1. Figure 7 shows that the token load across text experts is highly unbalanced. Do these experts correspond to particular semantic groups or patterns? Could you provide further analysis, and would introducing a load-balancing loss help?

2. In the SVA module, the image-to-text alignment require computing affinities of experts. How is this objective applied or adapted to the baselines reported in Table 2?

3. Table 8 reports efficiency analysis on UT-Zappos, but this dataset has relatively few state and object categories. Since the efficiency is sensitive to the number of primitives, could you report or discuss results on other datasets and under open-world settings?

4. Could you provide a more detailed analysis of the vision experts? For example, showing how the same expert allocates patches across different images?

5. It would be valuable to include a more detailed investigation of text experts. For instance, analyzing the impact of using separate experts for states, objects, and compositions.

6. Which dataset was used for the qualitative analyses in Figures 7–8?

---

> ### Author Response · Authors · 2025-11-24
>
> We sincerely thank the reviewer for the positive assessment of the soundness, presentation quality, and contributions, as well as for recognizing the effectiveness of EVA’s MoE-based design and the SVA alignment mechanism. Below we address the weaknesses and questions in detail.
>
> **Q1. Token load imbalance and ablation study on the number of experts.**
>
> **A1.** Yes, from the perspective of token load, the experts are unbalanced and we didn’t utilize the load balancing loss. We think that MoE routing imbalance in EVA is not a defect but a reflection of the intrinsic heterogeneity of primitive semantics. There are primarily two reasons for this:
>
> (1). The purpose of introducing expert learning is to achieve semantic differentiation, and artificially balancing the load would contradict this intention. This reflects **natural specialization** rather than problematic imbalance, as shown in Fig 4 and 5.
>
> (2). Load balancing encourages experts to be used uniformly, which contradicts the natural semantic distribution of primitives: Primitive semantics are not uniformly distributed across contexts (“old” appears more often with people than fruits), and forcing equal expert routing pushes EVA toward homogenized experts, reducing specialization.
>
> To validate it, we designed a controlled experiment, and the results are as follows:
>
> | Method | U | S | AUC | HM |
> | :--- | :--- | :--- | :--- | :--- |
> | **Eva** | 44.6 | 47.1 | 18.8 | 36.9 |
> | **+Load Balance** | 44.2 | 46.8 | 18.5 | 36.4 |
>
>
> We observe a consistent but small performance drop when applying a load-balancing (LB) loss.
>
> | Expert Num | U | S | AUC | HM |
> | :--- | :--- | :--- | :--- | :--- |
> | **1+6** | 44.0 | 46.5 | 18.2 | 36.2 |
> | **1+8** | 44.6 | 47.1 | 18.8 | 36.9 |
> | **1+10** | 44.0 | 46.0 | 18.0 | 36.0 |
>
> It demonstrates that the model with 8 experts achieves the highest performance across all metrics, highlighting that a moderate level of expert diversity is most effective for capturing semantic heterogeneity and enabling compositional reasoning. Models with too few experts tend to underfit the complex variations in semantics, failing to disentangle meaningful components, whereas models with an excessive number of experts suffer from routing sparsity, which reduces the effectiveness of each expert and weakens overall specialization. This indicates that balancing the number of experts is crucial for achieving both expressive power and efficient specialization in semantic decomposition.
>
> &nbsp;
>
> **Q2. SVA image-to-text affinity.**
>
> **A2.** Sorry for this confusion. The idea of SVA is considering the sematic diversity within states and objects. To evaluate the effect of SVA, we only implement the MoE adaptors in the final layer of image/text encoders (Baseline), while Domain-expert Adaption injects MoE adaptors in all layers.
>
> &nbsp;
>
> **Q3. Efficiency analysis on UT-Zappos only — can results on larger datasets or open-world settings be discussed?**
>
> **A3.** Thank you for pointing this out. We chose UT-Zappos because it is widely used in previous works for efficiency analysis,
>
> To address your concern, we have now profiled MIT-States dataset in closed-world setting. The results are tested on a A800 GPU and the batch size is set to 128.
>
> | Model | #Trainable | Training Time (one epoch) | Inference Time | AUC |
> | :--- | :--- | :--- | :--- | :--- |
> | **DFSP** | 43.3M | 3min45s | 17.2ms | 20.6 |
> | **Ours** | 41.7M | 3min20s | 17ms | 24.0 |
>
> Our approach achieves primitive polysemy learning through a structurally multi-expert design, so the number of primitives does not affect efficiency. In summary, we achieve superior performance and inference speed with fewer trainable parameters.
>
> &nbsp;
>
> **Q4. Additional expert-level visual analysis: same expert across different images.**
>
> **A4.** We agree this is valuable. **In Figure 6, we present the distribution of visual experts across three scenarios.** We observe that different experts focus on different subjects and the objects handled by the same expert in different scenarios exhibit a certain degree of similarity. For example, Expert 3 is more interested in “purple” object and Expert 7 has a high affinity for vegetables. Considering the limited number of experts and the complexity of semantic compositions, different experts do not exhibit clear semantic boundaries.

---

> > ### Author Response · Authors · 2025-11-24
> >
> > **Q5. Separate experts for states, objects, and compositions.**
> >
> > **A5.** In order to further study the impact of semantic isolation on multi-expert learning, we also conducted the comparison experiment. Specifically, we use two identical MoE adaptors for state and object, respectively. Each adaptor with one share expert and eight routed experts has same structure with EVA. We utilized the most complex dataset C-GQA and results are shown：
> >
> > | Method | U | S | AUC | HM |
> > | :--- | :--- | :--- | :--- | :--- |
> > | **EVA** | 44.6 | 47.1 | 18.8 | 36.9 |
> > | **EVA + semantic isolation** | 44.2 | 46.7 | 18.5 | 36.4 |
> > | **EVA + semantic isolation + token balance** | 43.9 | 46.3 | 18.0 | 36.0 |
> >
> > Introducing *semantic isolation* results in a slight performance degradation. This suggests that strictly isolating experts doesn’t improve performance on compositional generalization. It might be the mixture of attributes and objects facilitates experts in learning semantic relationships, while arbitrary divisions offer no benefit to the model. This is also similar to the current situation of MoE-based LLMs. Considering the impact of imbalance between experts, we also introduce a balancing loss to alleviate it. It is observed that balanced token load also performs worse than EVA. This further demonstrates that in compositional zero-shot tasks, the limited semantic content makes expert adaptive learning more effective.
> >
> > We will report these results in the revision.
> >
> > &nbsp;
> >
> > **Q6. Dataset used for Figures 7–8.**
> >
> > **A6.** MIT-States was used for Figures 7–8.

---

> ### Author Response · Authors · 2025-11-27
>
> Dear Reviewer BePU,
>
> We are delighted that you recognize the value of our work and hope it brings novel insights to CZSL. Your questions will greatly assist us in refining this breakthrough work!
>
> We deeply appreciate your feedback, particularly regarding the model analysis and further discussion with experts. We kindly request your time to review our rebuttal, which we believe addresses all your concerns. We look forward to continuing our discussion. Wishing you all the best!

---

### Official Review · Reviewer_8hsj · 2025-11-01

**Soundness:** 3
**Presentation:** 3
**Contribution:** 2
**Rating:** 4
**Confidence:** 4

**Summary:**

This paper proposes a mixture-of-experts framework for semantic variant alignment (EVA) to solve compositional zero-shot learning. The proposed method introduces two main modules: domain-expert adaptation and semantic variant alignment. The extensive experiments on three benchmarks demonstrate significant improvements over previous SOTA.

**Strengths:**

1) This paper introduces MoE paradigm to model the heterogeneous nature of primitive concepts.
2) The experimental results are promising.

**Weaknesses:**

1) The novelty is incremental. Specifically, many CZSL methods enable the primitive embeddings to dynamically adapt to diverse semantic variants by fine-grained learning[3], clustering-based prototypes[1], distribution learning[2] et al. The introduction of MoE to facilitate the adaptation of features is sound but this is merely a simple application without promoting the development of CZSL.
2) Some parts of the paper are difficult to understand. Terms like "domain-expert adaption," “in-domain knowledge”, "variant-based method" are used with little prior explanation.
2) The inclusion of a balancing coefficient $\alpha$ is also empirically motivated rather than theoretically justified, and it is unclear how sensitive results are to this choice.
3) In lines 258-269, the feature variants V is not described clearly, and the selection of state/object image features from feature variants lacks rigorous mathematical support or analysis. In addition, A_S \in R^{N_e+1}, why i\in [0,N_e] in equation (11)?
4) In lines 267-268, is {v_i}_{i=0}^{N_e} feature variants or semantic variants?
5) In lines 262-263, why is A_v computed based on V and image features f_c, while A_s and A_o are computed based on V and text features t_s/t_o?
[1] LEARNING CLUSTERING-BASED PROTOTYPES FOR COMPOSITIONAL ZERO-SHOT LEARNING
[2] Prompting Language-Informed Distribution for Compositional Zero-Shot Learning
[3] Leveraging sub-class discimination for compositional zero-shot learning

**Questions:**

Addressing weaknesses and improving writing quality.

---

> ### Author Response · Authors · 2025-11-24
>
> We sincerely thank reviewer for the constructive feedback and recognizing the strengths of our work. Below we address all concerns point-by-point and clarify misunderstandings.
>
> **Q1. The novelty is incremental.**
>
> **A1.** Sorry for this confusion. Please allow us to clarify that our contribution goes beyond an incremental use of MoE.
>
> **First**, contextuality caused by primitive polysemy is one of the focuses of Compositional Zero-Shot Learning(CZSL). Prior CZSL methods—whether based on fine-grained prompt adaptation[1], clustering[2], or distribution modeling[3]—try to improve the diversity of features, and address the entanglement within state-object pair. Specifically, PLID[1] formulates the language-informed class distributions by leveraging pre-trained large language models (LLM), which affects deployment costs and operating speed. CLUSPRO[2] requires additional space to maintain a set of diversified prototypes for modeling the diversities of primitives. Hu[3] defines the primitive concepts in different compositions as sub-classes, maintaining the sub-class discrimination. These above excellent works all introduces existing (not “novel”) methods(clustering) or modules(LLM) to solve the same problem in CZSL and inspire subsequent research. This proves that new effective insights achieved by appropriate models are meaningful to CZSL.
>
> **Second**, in this paper EVA proposes a novel insight to reformulate primitive polysemy as semantic heterogeneity and introduces an **architectural decomposition** where multiple experts encode distinct semantic subspaces of the same primitive, enabling context-aware semantic variant modeling. Furthermore, our **semantic variant alignment** is a new cross-modal mechanism: it considers semantic polysemy and performs bidirectional global-to-local variant selection on both text and image sides, rather than the one-to-all primitive alignment used in prior CZSL works. This introduces a new alignment paradigm and yields substantial gains, far exceeding typical incremental improvements in CZSL.
>
> &nbsp;
>
> **Q2. Some parts are difficult to understand (domain-expert adaption / in-domain knowledge / variant-based method).**
>
> **A2.** We appreciate the reviewer's point and have revised the original paper presentation to improve its clarity. In the revision: 1. **Domain-expert adaptation** will be introduced as: “using MoE adapters to dynamically assign tokens to specialized experts for modeling heterogeneous primitive semantics.” 2. **In-domain knowledge** refers to the specific information associated with tokens routed to the same expert (e.g., color). 3. **Variant-based method** will be clarified as: “constructing multiple semantic variants (text or image) and selecting the most semantically relevant one for primitive alignment.”
>
> &nbsp;
>
> **Q3. The balancing coefficient α.**
>
> **A3.** Sorry for this confusion. The coefficient α is designed to balance inter- and intra-model affinity. Similar to the balancing coefficients in loss functions, we aim to comprehensively consider the two types of affinity in order to select appropriate feature variants. We add the **sensitivity analysis of α** in the revision:
>
> | α | U | S | AUC | HM |
> | :--- | :--- | :--- | :--- | :--- |
> | **0** | 43.8 | 46.2 | 18.0 | 35.8 |
> | **0.3** | 44.2 | 46.9 | 18.6 | 36.6 |
> | **0.5** | 44.6 | 47.1 | 18.8 | 36.9 |
> | **0.7** | 44.0 | 46.8 | 18.5 | 36.3 |
> | **1** | 43.8 | 46.8 | 18.2 | 36.0 |
>
> Across the range from \$\\alpha = 0\$ to \$\\alpha = 1\$, all metrics vary only slightly, indicating that EVA is largely insensitive to the exact choice of \$\\alpha\$. Performance remains stable over a broad interval, suggesting that the model is robust to this hyperparameter.
>
> This has been added in appendix.
>
> &nbsp;
>
> **Q4. Feature variants V not clearly described**
>
> **A4.** We apologize for the insufficient explanation. In the revision: 1. V = {vᵢ}₀ⁿᵉ is the set of CLS outputs from each expert in the last encoder layer, representing image feature variants from different semantic subspaces. 2. Index i ∈ [0, Ne] includes the shared expert (i=0) and Ne routed experts.
>
> &nbsp;
>
> **Q5.  Are v\_i feature variants or semantic variants?**
>
> **A5.** They are image-side **feature variants**, produced by different experts.
>
> &nbsp;
>
> **Q6.Why is A\_v computed using V and f\_c, while A\_s/A\_o use V and t\_s/t\_o?**
>
> **A6.** The motivation: 1. A\_s, A\_o (inter-model affinity) ensure the selected image feature variant matches the primitive(state, object) semantics in text. 2. A\_v (intra-model affinity) ensures the selected image feature variant aligns with the main visual semantics (global CLS feature). This combination prevents selecting variants too far from the image’s overall content.
>
> [1] Leveraging sub-class discimination for compositional zero-shot learning
>
> [2] Learning Clustering-based Prototypes for Compositional Zero-shot Learning
>
> [3] Prompting Language-Informed Distribution for Compositional Zero-Shot Learning

---

> ### Author Response · Authors · 2025-11-27
>
> Dear Reviewer 8hsj,
>
> We apologize for bothering you during your busy schedule. We believe you value this work and hope to collaborate with us in introducing the outstanding MOE mechanism to the CZSL field. The questions you raised will also greatly assist us in advancing this work toward the long-term direction we both envision.
>
> We greatly value your comments, particularly regarding methodological innovation and the description of certain details. We hope you will take the time to review our rebuttal, which we believe addresses all your concerns. We look forward to further discussion. Wishing you all the best!

---

### Comment · Area_Chair_gA4T · 2025-11-24

Dear Reviewers,

Despite there is no rebuttals from the authors, **we still kindly encourage you to read other reviewers' comments and revise your ratings, if need**. Your timely feedback is important for ensuring a fair and thorough review process. Thank you for your contributions to ICLR 2026.

AC

---

> ### Author Response · Authors · 2025-11-24
>
> Dear AC and Reviewers,
>
> We apologize for the delay in responding. Over the past period, we have been supplementing experiments, addressing the rebuttal points, and discussing improvements to our work. We have just completed this work and will submit our rebuttal shortly.
>
> We apologize for any inconvenience caused. However, I am confident that both we and the reviewers and AC share the common goal of contributing to the publication of this excellent work.
>
> Sincerely,
> The Authors

---

### Author Response · Authors · 2025-11-27

We sincerely thank all reviewers for dedicating their valuable time, exerting considerable effort, and providing insightful feedback. We have made every effort to address all raised concerns and engaged in more in-depth discussions where necessary. In responding to each reviewer, we carefully considered all points of concern and supplemented relevant experiments wherever possible.

We sincerely thank the reviewers who highly valued our work. We hope more reviewers will recognize the extent of our work and our rigorous approach. Should you have any further questions regarding our work, please do not hesitate to contact us. We are eager to engage in in-depth discussions and learning with experts in the CZSL field.

***We would be most grateful if reviewers could reassess their ratings based on the revisions we have implemented.***

Sincerely, The Authors

---

### Author Response · Authors · 2025-12-01
**Comprehensively addressed the issues and concerns**

We sincerely thank all reviewers for taking the time to thoroughly evaluate our manuscript. We have revised the paper based on your comments (revisions highlighted in blue) and provided comprehensive responses. We believe we have addressed all concerns raised by the reviewers:

1, We have restated the innovation and unique insights of EVA based on the comments from reviewers 8hsj and Fu3k, demonstrating that our work far surpasses the typical incremental improvements seen in CZSL.

2, We have revised the presentation to improve clarity, specifically redefining terms like "domain-expert adaptation," "in-domain knowledge," and "variant-based method," according to Reviewer 8hsj's comments.

3, We added a sensitivity analysis of the balancing coefficient $\alpha$ to demonstrate the model's robustness, according to Reviewer 8hsj's comments in the Appendix.

4, We clarified the definition of feature variants $V$ and the index range in Equation (11), according to Reviewer 8hsj's comments.

5, We added ablation studies regarding the number of experts and the impact of load balancing loss to validate our design choices, according to Reviewer BePU's, Reviewer Fu3k's, and Reviewer nDeF's comments.

6, We expanded the efficiency analysis to include the MIT-States dataset in the closed-world setting, demonstrating our method's superior balance between performance and computational cost, according to Reviewer BePU's and Reviewer Fu3k's comments in the Appendix.

7, We provided additional visualizations and analyses of visual expert distributions (Fig. 6) and semantic variants (Appendix Fig. 10) to enhance interpretability, according to Reviewer BePU's and Reviewer nDeF's comments.

8, We added comparative experiments on semantic isolation (using separate experts for states and objects) to justify our shared expert design, according to Reviewer BePU's comments.

9, We reorganized the figures and revised the descriptions in Sections 3.1–3.5 to reduce repetition and improve flow, according to Reviewer Fu3k's comments.

10, We added experiments replacing the CLIP backbone with SigLIP-B-16-224 to demonstrate the generalization of our framework across different vision-language models, according to Reviewer Fu3k's and Reviewer nDeF's comments.

We have made every effort to address each issue and concern, and we believe this work will advance progress in the field.

Sincerely yours, Authors.

---

### Author Response · Authors · 2025-12-01
**Author Final Remarks**

We sincerely thank AC and all reviewers for their valuable time and effort dedicated to this research. We have greatly benefited from the review process. The reviewers' insightful and constructive comments not only helped us identify and improve shortcomings in the paper but also inspired us to consider the problem from more diverse perspectives.

Here, I would like to briefly review our main contributions. This paper proposes the EVA framework to address **primitive polysemy and composition divergence** in the CZSL:

- **Introduces MoE for fine-grained representation**: We pioneer the use of MoE Adapter for Domain-Expert Adaptation in CZSL. Through a dynamic collaborative mechanism of “shared experts + routing experts,” we achieve token-aware, fine-grained primitive representation learning, overcoming the expressive limitations of static prototypes.
- **Proposing the Semantic Variant Alignment mechanism**: A dual-perspective cross-modal alignment strategy was designed. By dynamically selecting the most relevant image/text semantic variants, adaptive fine-grained semantic matching is achieved, significantly enhancing the model's ability to capture complex semantics.
- **Excellent Performance**: We comprehensively surpass existing SOTA on three major benchmark datasets. Particularly outstanding in open-world settings (AUC +2.6% on C-GQA), this fully validates the effectiveness of MoE routing and variant alignment mechanisms in enhancing compositional generalization. Comprehensive ablation studies and discussions substantiate our innovations and insights.

We believe EVA offers a novel CZSL solution grounded in the MoE framework, balancing expressiveness and generalization capabilities, and provides novel insights for visual-language modeling.

Guided by our shared objectives, we have carefully revised the paper based on reviewer feedback to **achieve higher academic standards** in theoretical analysis, model discussion, and results presentation. We are committed to further refining this work to ensure its scientific rigor and believe it meets the high standards of ICLR, a premier conference in the machine learning.

Once again, we extend our heartfelt gratitude to the AC and all reviewers for their professionalism and scholarly contributions. **We firmly believe that true gold fears no fire, and it is through our collective efforts and discussions that academic research can continually advance and benefit a broader research community.**

---

### Meta-Review · Area_Chair_MPR1 · 2026-01-06

**Summary:**

Reviewers’ concerns: Novelty (perceived incremental), unclear terms, lack of sensitivity analysis, vague feature variants description, insufficient ablation studies (expert number, load balancing), unclear figures, limited efficiency analysis, CLIP dependency, inadequate interpretability, expert imbalance/routing stability. Authors fully addressed all via clarifying innovation, revising terms, adding analyses/experiments, expanding efficiency data, validating across backbones, and enhancing visualizations.

**Reviewer Concerns:**

Seems all concerns are addressed. No outstanding ones.

**Reviewer Scores:**

no discussion.
Reviewer 8hsj, nDeF could change their score as most of the concerns are answered by the authors with care.

---

### Decision · Program_Chairs · 2026-01-26

Accept (Poster)